# Simulation Analysis of Delamination Damage for the Thick-Walled Composite-Overwrapped Pressure Vessels

**DOI:** 10.3390/ma15196880

**Published:** 2022-10-03

**Authors:** Houcheng Fang, Di Wang

**Affiliations:** Department of Mechanical Engineering, Xinjiang University, Urumqi 830017, China

**Keywords:** finite element analysis, delamination damage, tiebreak contact algorithm

## Abstract

In order to verify the delamination damage occurring in thick-walled composite-overwrapped pressure vessels, firstly, for composite delamination damage, a composite laminate model was established. Model I and model II delamination failure processes of composite structures were simulated and verified based on a tiebreak contact algorithm for different mesh sizes, respectively, and the approximate equivalent results were achieved by correcting the inter-ply strength. Then, for in-plane damage to composite materials, the elastic–plastic process was verified by selecting a progressive damage model, with quasistatic nonlinear tensile shear of sample specimens as an example. Further, under the purpose of generality and simplicity, the location of the first occurrence of delamination failure was simulated and analyzed with the tiebreak contact algorithm and a reasonable mesh size, using quasistatic loading of a thick composite-overwrapped pressure vessel cylindrical section as an example. The results showed that delamination occurred at approximately the center, which is in general agreement with the experimentally observed phenomenon. On this basis, the locations of the first significant delamination phenomena in composite-overwrapped vessels under three different ratios of plus or minus 45-degree layup angles were predicted. Finally, the differences in structural strength between the single laying methods and the combined laying method were compared. The results showed that the ratio of 50% had a higher modulus value than a pure 0° ply, but too large a ratio was detrimental to the improvement of structural properties.

## 1. Introduction

Because of its abundant supply, low pollution, and high calorific value, hydrogen energy has emerged as a significant answer to the energy problem and to achieving the “double carbon” target. Hydrogen-powered vehicles have emerged as the future vehicle trend [1,2,3]. A composite-overwrapped pressure vessel is a fiber-wrapped composite construction with many layers. These tanks are typically used for high-pressure hydrogen gas storage [4]. Composite materials used in composite-overwrapped pressure vessels are preferred over traditional materials due to their higher specific strength, stiffness, and fatigue resistance. However, the interlaminar fracture strength of composite materials is poor, and the incidence and extension of delamination significantly lower a structure’s strength and stiffness, posing a threat to the structure’s integrity and safety [5].

The cohesion zone model (CZM) has been widely used in the simulation of delamination failure of composite structures in recent years [6]. The main simulation methods to describe the delamination failure of composite structures are the cohesive element method and the tiebreak algorithm based on the cohesive contact method, etc.

Song, L. A., et al., used a progressive damage model based on the Puck criterion to model matrix fracture and fiber fracture (in-plane failure) in composite pressure vessels. Cohesive elements were inserted between pressure vessel layers to model interlayer failure. The simulation results were consistent with the experimental results, and the key parameters for the progressive damage analysis of composite pressure vessels were determined [7]. Liao, B., et al., embedded bilinear cohesive elements in a composite pressure vessel model and investigated the evolution of interlaminar damage under low-velocity impact loading evolution. More severe delamination damage occurred in composite pressure vessels with aluminum lining compared to EPDM lining [8]. Weerts et al., prelaminated a hydrogen storage tank and simulated delamination failure of the column section under quasistatic loading conditions by inserting zero-thickness cohesive elements. Interlaminar failure was usually the first visible mechanism for thick rings, whereas fiber fracture prevailed for thin rings [9]. Aleksandr Cherniaev et al., used the tiebreak contact algorithm to model interlaminar damage behavior and analyzed the effect of nonphysical parameters of the interlaminar material model on the prediction of compressive damage loads in damaged composite cylinders. The peak stress parameter NLFS of the interlaminar model was able to accurately predict the damage loads observed in physical experiments when the parameter SLIMC, associated with the continuous damage criterion, was between 0.6 and 1 [10]. Jinyang Zheng et al., used the tiebreak contact algorithm to simulate interlaminar failure behavior. The amplitude and phase inconsistency between adjacent layers of composite materials during vibration under explosive loading was the main cause of delamination failure [11]. Montes De Oca Valle developed a finite element model of a specimen with less complexity than a whole cylinder. An explicit solution was used to predict the delamination damage between the composites and the separation of the metal from the composite by inserting cohesive elements. Finally, the metal–composite interface properties were obtained and calibrated through experimental tests [12]. Drew E. Sommer et al., conducted an explicit finite element analysis of the damage response of a broken carbon fiber composite tube impacted by a falling hammer. A continuous damage mechanic (CDM) model was used to simulate interlaminar damage to the composite, and a tiebreak contact algorithm was used to simulate delamination damage to the composite. The interaction between delamination fracture energy and friction could greatly affect the results [13]. These researchers have thoroughly investigated the external construction of composite columnar containers but have not evaluated the influence of different lying angles on the inside.

George Edward Street et al., investigated the effect of preload on large composite structures. When the loading was not parallel to the major fiber direction, the effect of preload forces could be severe. Matrix cracking and delamination could be more severe under low-velocity impact loading due to the effect of preload. The relationship between the loading main fiber direction and angle was analyzed. When the two were not parallel, the preloading had a greater impact [14]. Naseer H. Farhood et al., predicted the first ply failure pressure based on finite element simulations of Tsai-Wu and maximum stress failure criteria. The effects of ply stacking order and orientation angle on the burst pressure were investigated with a constant ply thickness PV. However, the layering method was simple and did not separately consider the effect of different angular proportions [15]. Zhengyun Hu et al., designed twelve stacking schemes for Type IV hydrogen storage vessels and evaluated the effect of different stacking sequences on the ultimate strength. The effect of the lamination order on the rupture pressure of the hydrogen vessel was about 15%. The laying sequence of the separated hoop and helical layers could increase the bursting pressure [16]. Salamat-Talab et al., prepared and tested laminated composites with 0//θ (θ = 0, 15, 45, 60, and 90) interfaces, respectively, and obtained Mode II laminate fracture toughness and cohesive shear traction–separation models. The experimental results showed that the crack extension behavior in the specimens became more stable with the increase in the interfacial fiber angle [17]. At the moment, research on the effect of varied laying orientations on composite outsourcing pressure vessels is insufficient. Few researchers have investigated the influence of a single laying method on structural strength, particularly the influence of a laying ratio of plus or minus 45 degrees on the structural strength of a thick-walled composite cylinder.

In this paper, a method of simulating the delamination of Model I and Model II composites using the tiebreak contact algorithm under different mesh sizes is verified by modifying the interlaminar strength. Additionally, under the verification of this method, a quasistatic loading simulation is carried out on the cylinder part of a thick-walled composite-overwrapped pressure vessel with a reasonable mesh size, and the location of the first significant delamination phenomenon of the thick-walled composite-overwrapped pressure vessel is verified. By increasing the proportion of spiral-wound wires (plus or minus 45 degrees) in the polar-wound layup, the effect of different plus or minus 45 degrees layup ratios on the structural strength of the compression ring is analyzed. In comparison, the fibers laid alternately at plus or minus 45 degrees can provide better structural strength in the circumferential and radial directions.

## 2. Problem Description

In a cohesive model, the amount of stress separation in the cohesive model artificially reduces the overall stiffness of the material due to the presence of the initial elastic phase. It is well-suited to describe the delamination phenomenon between layers in composite materials.

Although a CZM has requirements on the model mesh size, for larger simulation structures, larger mesh sizes can be used appropriately to save computation time when similar results can be achieved by correcting the interlayer strength. The tiebreak contact algorithm differs from the cohesive element method in that the surface-based cohesive behavior is a contact property rather than a material property. The tiebreak contact algorithm does not involve the usage of building finite elements, but rather exploits the nodes between the upper and lower sublayer elements to mimic interlayer bonding, as illustrated in Figure 1 [18]. In solving interface problems, the advantages of cohesive behavior over cohesive elements are as follows: no extra cohesive elements are required, the number of participating elements is reduced, and the results are relatively easier to converge.

To represent delamination, the two contact interfaces between the TSHELL element layers are modeled using *CONTACT_AUTOMATIC_ONE_WAY_SURFACE_TO_SURFACE_TIEBREAK_{OPTION} with OPTION = 9. OPTION = 9 is based on the fracture model in the viscous material model *MAT_COHESIVE_MIXED_MODE. The principle is shown in Equation (1):(1)$${\left(\frac{\left|\sigma_{n}\right|}{\mathrm{NFLS}}\right)}^{2}+{\left(\frac{\left|\sigma_{S}\right|}{\mathrm{SFLS}}\right)}^{2}\geq 1$$
where $\sigma_{n}$ is the normal stress, $\sigma_{S}$ is the shear stress, NFLS is the normal interlaminar strength, and SFLS is the shear interlaminar strength [19]. In the load–displacement curve, this option is feasible for obtaining similar results with coarse meshes as with fine meshes. However, coarser meshes require lower peak traction forces [20]. As the load keeps increasing, the stress σ between the layers of the specimen continues to increase; when the stress σ reaches the failure criterion, the contact point starts to break down, and the interface starts to separate. After that, as the interface separation distance δ decreases linearly, the interface separates further until it reaches the interface limit separation distance, the stress σ decreases to 0, the contact point is completely destroyed, and the interface separates completely, marking the beginning of crack generation. With the complete destruction of contact between multiple groups of nodes, the crack starts to expand. The energy release rate at the interface separation is, theoretically, equal to the area of the region below the σ–δ curve, as shown in Figure 2.

In order to obtain accurate calculation results, the model needs to be divided into a fine grid, with the length of the layered process area to be divided into a certain number of elements to meet the requirements of Equation (2):(2)$$N_{e}=\frac{L_{\mathrm{cz}}}{L_{e}}$$
where $L_{\mathrm{cz}}$ is the length of the cohesive zone, $L_{e}$ is the length of a single element on the crack extension surface, and $N_{e}$ is the number of elements to be divided. Turon et al., showed that it was not less than 3. That is, the size of the elements in the cracking direction was not larger than 0.5 mm to ensure the accuracy of the calculation [21]. This severely limits the application of the tiebreak algorithm in large-scale simulations. Rice et al., proposed a series of cohesive models where the definition of the length of the stratified process area could all be expressed uniformly in Equation (3) [22]:(3)$$L_{\mathrm{cz}}=\mathrm{ME}\frac{G_{c}}{{\left(\tau^{0}\right)}^{2}}$$
where E is the Young’s modulus of elasticity of the material; $G_{c}$ is the critical energy release rate; $\tau^{0}$ is the interlaminar strength; and *M* is a constant less than 1, where the value of M has not been determined. The most commonly used values are Rice [22] and Hillerborg [23], which are considered to be 0.88 or 1. Pedro PC et al.’s, study showed that the accuracy of the simulation results was mainly affected by the energy release rate $G_{c}$, and the correction of the interlaminar strength could be ensured by the constant energy release rate ${G_{c}}^{\prime }$ (as shown in Figure 2). The simulation accuracy could be guaranteed when the two areas were equal. It can be seen from Equation (2) that, when the interlaminar strength value is reduced, the length of the cohesive zone becomes larger. Under the condition that the number of grids in the cohesion zone is sufficient, a larger grid size can be used to quickly and accurately calculate the damage and failure load of the adhesive layer, thereby improving the calculation efficiency. This provides the possibility to simulate composite delamination using the tiebreak algorithm under meshes of different sizes.

## 3. Simulation of Model I and Model II Delamination Failure of Composite Materials

In order to model delamination failure appropriately, two delamination modes were considered to be relevant within the vessel model: BCD (opening delamination) and ENF (shear forward delamination) [24]. The tiebreak contact algorithm determined the failure equation of the interface from two main types of interlaminar fracture modes, model I and model II, and could accurately simulate the crack generation and expansion process. Since the test methods for pure model I and model II fracture toughness have become mature, the fracture toughness ($G_{Ic}$, $G_{IIc}$) in the model I and model II modes is usually selected as the basic parameter to describe the failure response of mixed-mode composite structures, as shown in Figure 3.

The specimen material parameters were selected based on the literature [25]. The specimen length was 100 mm, the width was 20.0 mm, the height was 2 h, each h thickness was 1.55 mm, and the initial delamination length was 35 mm. In the finite element model calculation, the model elements model was selected as TSHELL elements for the model I and model II delamination failure simulations, with element lengths of 0.4 mm, 0.5 mm, 1.0 mm, 1.5 mm, and 2.0 mm. The total numbers of elements were 100, 400, 64,000, 8160, 1742, and 1020, as shown in Figure 4.

### 3.1. DCB Delamination Failure Simulations

In the model Ⅰ DCB test, the energy release rate versus time is shown in Figure 5, which shows that $G_{Ic}$ played a major role, and the effect of $G_{IIc}$ on delamination was negligible. The calculation results of the displacement of the load with the load application point for different element sizes are given in Figure 6.

The calculation results for element lengths of 0.4 mm and 0.5 mm were overlapped. An element length of less than 0.5 mm did not affect the calculation accuracy. The calculation results of the element length of 1.00 mm was very close to the calculation results of 0.4 mm and 0.5 mm. Figure 6a shows the uncorrected DCB load–displacement curves based on the tiebreak contact algorithm, from which it can be seen that the slopes of the load–displacement curves remained in good agreement for different grid sizes for the model I DCB test. When $L_{\mathrm{cz}}$ and $\tau^{0}$ were kept constant, the greater the mesh size, the greater the stress required for interface separation and the more noticeable the ensuing oscillation in delamination damage. Figure 6b depicts the load–displacement curve based on the tiebreak contact correction.

It can be seen that the different mesh densities that could be obtained after the conversion of interlayer strength could be very approximate. However, with increases in mesh size and length, the oscillation in delamination damage was still obvious. For an element length of 1.0 mm, the load–displacement curve based on the tiebreak contact algorithm agreed well with the load–displacement curves of 0.4 mm and 0.5 mm. The load–displacement curve for the element length of 1.0 mm ensured the accuracy of the calculation and had better calculation efficiency compared with the element lengths of 0.4 mm and 0.5 mm, Figure 7.

### 3.2. ENF Delamination Failure Simulations

The energy release rate versus time in the model II ENF test is shown in Figure 8, which shows that $G_{IIc}$ played a major role, and the effect of $G_{Ic}$ on delamination was negligible. The results of load and displacement calculations for the loading of members with different element lengths are given in Figure 9. Element lengths less than 0.5 mm did not affect the calculation accuracy. The calculation results for the element length of 1.00 mm were also very close to the calculation results of 0.4 mm and 0.5 mm. Figure 9a shows the uncorrected ENF load–displacement curve based on the tiebreak contact algorithm, from which it can be seen that the slope of the load–displacement curve also maintained a good agreement for different grid sizes for the model II ENF test. When $L_{\mathrm{cz}}$ and $\tau^{0}$ were kept constant, the larger the mesh size, the larger the load required for interface separation. Compared with the model I DCB test, the oscillation in delamination damage by increasing mesh size was not obvious. The load–displacement curves based on the tiebreak contact correction in Figure 9b was obtained through relatively reducing interlaminar strength. It can be seen that, for the model II ENF test, the results could still be obtained very approximately after the conversion of interlayer strength for different grid densities. The comparison between the two figures shows that the load–displacement curve based on the tiebreak contact correction with an element length of 1.0 mm still had a good agreement with the calculation results of 0.5 mm while improving the calculation efficiency.

In both the model I DCB test and the model II ENF test, the error increased with the increase in element length. However, the results for the element length of 1.0 mm showed good similarity with element lengths of 0.5 mm and 0.4 mm. This is in agreement with the findings of Camanho et al., in the literature [26].

## 4. Simulation of Cylindrical Partial Delamination Failure of Composite Pressure Vessel

### 4.1. Selection of Materials for In-Plane Damage of Composite Materials

When a composite-overwrapped vessel is loaded, delamination failure predominates, and in-plane failure hardly occurs. At this time, the fiber is in the elastic–plastic phase, and a MAT_54 progressive damage model can ideally simulate the elastic–plastic phase of the composite [27].

For in-plane damage to composite materials, the nonlinear tensile shear of sample specimens was used as an example, and the relevant parameters and specimen dimensions of CFRP composites were obtained from the literature [28]. The length and width of the specimens were 200 mm and 25 mm, respectively, and the stacking order of the tensile specimens was ([±25]7T), which were simulated and modeled according to the international standard (ASTMD3039), as shown in Figure 10a. The model was modeled with a combination of TSHELL elements and COMPOSITE_TSHELL. The length of the elements was 1.00 mm. The total number of elements was 10,000, as shown in Figure 10b. The MAT_54 progressive damage model was used for the in-plane damage material. As long as the inertial forces were not considerable and the material model utilized was not rated as sensitive, the loading speed could be raised significantly to lower the needed computational time [29]. The fixed end and the loading end were set based on the real loading conditions, and the speed-loading method was used. The speed was set to 0.005 mm/ms to reduce computation time.

Figure 10 depicts the computed findings of the stress–strain curve of the specimen in quasistatic tension (b). The slope of the simulated numerical section in nonlinear shear was substantially equivalent to the experiment, and the stress grew uniformly with strain, which could well-describe the elastic phase of the tensile process. By adjusting the stiffness retention factor during the plastic phase, the stress could be kept constant at around 720 MPA. The material model could clearly respond to the elastic–plastic process of nonlinear shear in composite materials.

### 4.2. Model Construction of the Column Part of Composite Pressure Vessel

Since the physical tests [9] were performed under quasi-two-dimensional conditions, as shown in Figure 11a,b, a two-dimensional simplified modeling approach similar to that in the literature was used, and the vessel model was simulated using explicit analysis for quasistatic loading. As shown in Figure 11c, the model was built concerning the literature [9]. The model consisted of three parts: the indenter, the column part, and the support.

The indenter was moved vertically downward, while the support was entirely stationary. The deformation of the indenter and the support could be ignored to shorten the computational time of the simulation. These two components were represented by solid elements with a total of 6952 elements, and the material was MAT_ RIGID. The column section was short, with a length of 3 mm. The inner radius R_in_ was 150 mm, and the wall thickness was 25 mm. The front and rear surfaces were clamped in the longitudinal direction. The pressure vessel column section consisted of 20 CFRP layers, modeled by a combination of TSHELL elements and COMPOSITE_TSHELL. Each CFRP layer was represented by one element in thickness and three elements in length. To maintain calculation efficiency, the element length was limited to around 1 mm. There were 60,000 elements in total. The geometry of the CFRP layer model is depicted in detail in Figure 11d. The alternating blue and red components identify the CFRP layers.

The MAT_54 progressive damage model was selected for the in-plane damage material. The composite layers were bonded to each other with the tiebreak contact algorithm. The elastic material parameters used for the orthotropic anisotropic material were typical for CFRP and are shown in Table 1 [9]. Reference [30] proposed two types of CFRP interlayer strengths, “strong” and “weak”. Since increasing the elements’ length required a smaller interlayer strength, the difference between the 0.5 mm and 1 mm interlayer strength was not very large. Here, a more reasonable “weak” interlayer strength was chosen, as shown in Table 2.

### 4.3. Pressure Vessel Delamination Failure Analysis

In this paper, the same layup was used as specimen A in the literature [9] for the simulation analysis: the layup was [(±45°)_5_ /90°_2_]_24_. The comparison between experiments and simulations is shown in Figure 12. The simulated and experimental phenomena behaved consistently, with the first distinct stratification occurring approximately at the center. The trend of the curves showed a good correlation. The elastic response appeared first during loading. The end of the elastic response (point 1) was followed by a clear load drop (point 2), which was associated with the first delamination failure of the CFRP layers.

On this basis, the [0°]_100_ layups, the [(±45°)_5_ /0°_10_]_5_ layups, and the [±45°]_50s_ layups were studied respectively. Three different ratios of plus or minus 45-degree layups were considered and analyzed. To increase the ratio of spiral-wound (±45°) layups in polar-wound (0°) layups, the delamination effect is shown in Figure 13a–c. These CFRP layers consisted of a total of 20 layers. As can be seen in Table 3, regardless of the ratios of ±45° layups, the first delamination failure of the 25 mm thick ring occurred approximately in the center.

The order of occurrence of bending damage and shear damage during loading was thickness-dependent. For thick rings, the shear stress was greatest at the center, and shear damage occurred preferentially [9]. The cracks for interlaminar damage began to sprout in the center of the wall and then became visible. Since the loading conditions were approximated as two-dimensional conditions, the maximum values of radial stresses were distributed on both sides of the ring, as shown in Figure 13. Compared with the simplified [0°]_100_ layup model, the circumferential stresses, radial stresses, and axial stresses generated by the [(±45°)_5_ /0°_10_]_5_ layup model with significant delamination failure were larger than those of the simplified [0°]_100_ layup model. That is, the [(±45°)_5_ /0°_10_]_5_ layup model had better structural strength than the simplified [0°]_100_ layup model and was less prone to delamination failure when subjected to impact. The [±45°]_50s_ layup model required greater circumferential and radial stresses to achieve delamination failure than the [(±45°)_5_ /0°_10_]_5_ layup model, but the axial stress was less at this point. In conclusion, for thick-walled composite tanks, increasing the proportion of spiral-winding (plus or minus 45 degrees) layup in polar-winding layups could improve the structural strength of the structure in the radial and circumferential directions. However, for axial strength, the [(±45°)_5_ /0°_10_]_5_ combined layup model was higher than single polar winding [0°]_100_ or spiral winding [±45°]_50s_.

## 5. Conclusions

(1) In order to model delamination failure appropriately, a method of simulating the delamination of Model I and Model II composites using the tiebreak contact algorithm under different mesh sizes was verified by modifying the interlaminar strength. If the interlaminar strength σ remained constant as the displacement δ at delamination failure increased, the area of the energy release rate $G_{c}$ for delamination damage increased, and the load required for delamination damage increased with the corresponding increase. The oscillation caused by the delamination damage became more visible as well. If the interlaminar strength σ was reduced at this time, the area of the delamination damage’s energy release rate $G_{c}$ remained constant. The interlaminar strengths of various meshes were converted to produce very close results. Because of the fewer elements, the model with element lengths of 1.0 mm had higher computational efficiency than the models with element lengths of 0.4 mm and 0.5 mm, and the computational results were still in good agreement with those of the models with element lengths of 0.4 mm and 0.5 mm.

(2) When the thick-walled composite-overwrapped pressure vessels were quasistatically loaded, there was almost no in-plane failure, and the fibers were in the elastic–plastic stage. The elastic–plastic process was verified by selecting a progressive damage model with quasistatic nonlinear tensile shear of the material. The slope of the simulated numerical section was basically similar to that of the experiment, and the stress increased uniformly with strain, which could well-reflect the elastic phase of the tensile process. In the plastic phase, the stress could be kept constant at about 720 MPa by setting the stiffness retention factor. The progressive damage model could respond to the elastic–plastic phase of nonlinear shear of composite materials ideally.

(3) A 25 mm composite-overwrapped pressure vessel cylindrical model was established, and a quasistatic loading simulation was carried out. The first delamination failure occurred approximately in the center. In comparison, the fibers laid alternately at ±45° could provide better structural strength in the circumferential and radial directions. For axial strength, the [(±45°)_5_ /0°_10_]_5_ combined layup model was higher than single polar winding [0°]_100_ and spiral winding [±45°]_50s_. The ±45° layup had a direct effect on the flexural properties of the composite. The proportion of fibers at 50% had a higher modulus value than a pure 0° ply, and the structure could be stronger than a 0° ply in terms of load capacity after adding ±45° ply, but too large a ratio was detrimental to the improvement of structural properties.

This simulation study analyzed the difference in structural strength between the combined plus and minus 45-degree laying method and single laying methods, and specific ratio optimization will be the focus of future research.

## Figures and Tables

**Figure 1 materials-15-06880-f001:**
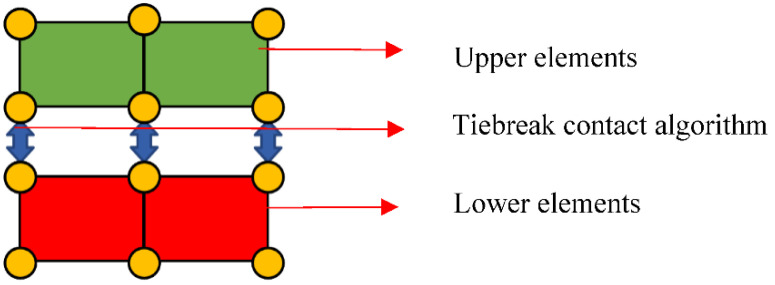
The tiebreak contact algorithm schematic diagram.

**Figure 2 materials-15-06880-f002:**
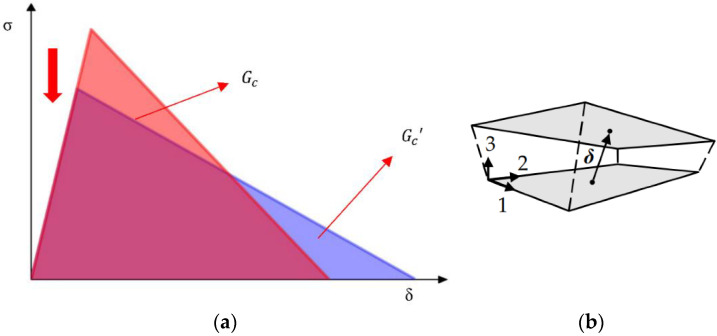
Relationship between energy release rate and interlaminar strength. (**a**) Energy release rate of the same area; (**b**) Parameters of the SURFACE_TO_SURFACE_TIEBREAK contact with OPTION 9.

**Figure 3 materials-15-06880-f003:**
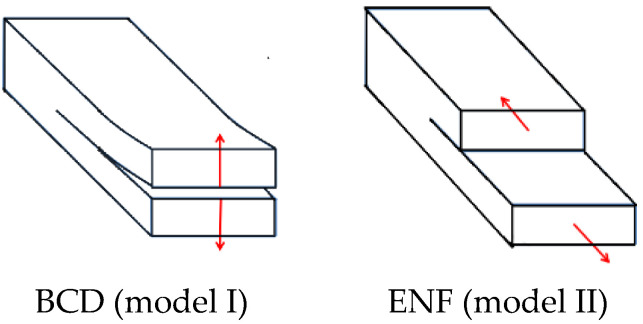
Example diagram of pure model I and pure model II delamination failure of composite materials.

**Figure 4 materials-15-06880-f004:**
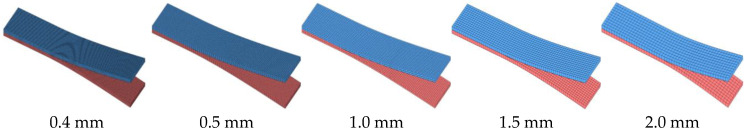
Model I with different element sizes.

**Figure 5 materials-15-06880-f005:**
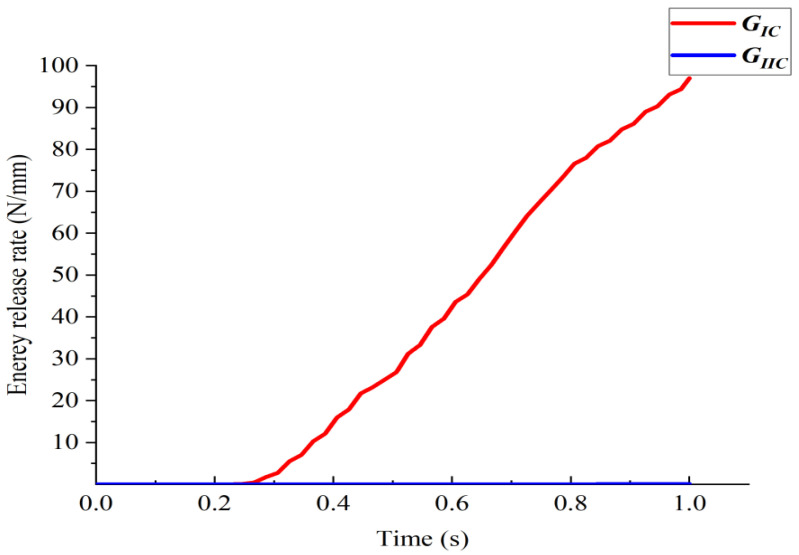
Energy release rate versus time for model I delamination failure.

**Figure 6 materials-15-06880-f006:**
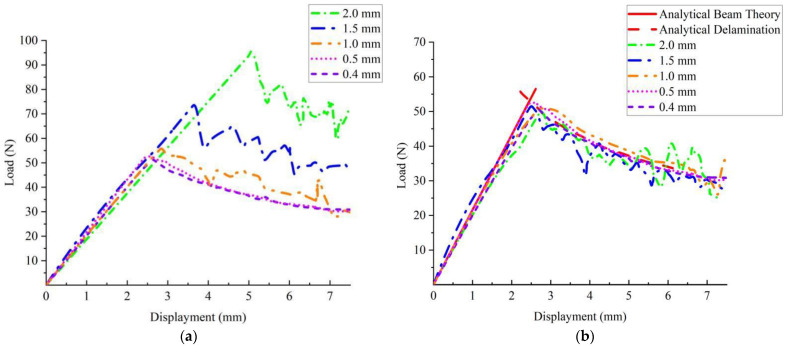
Model I delamination failure based on the tiebreak contact algorithm correction. (**a**) Uncorrected load–displacement curve; (**b**) corrected load–displacement curve.

**Figure 7 materials-15-06880-f007:**
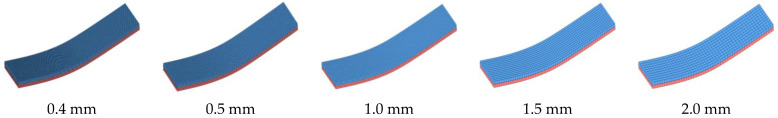
Model II with different element sizes.

**Figure 8 materials-15-06880-f008:**
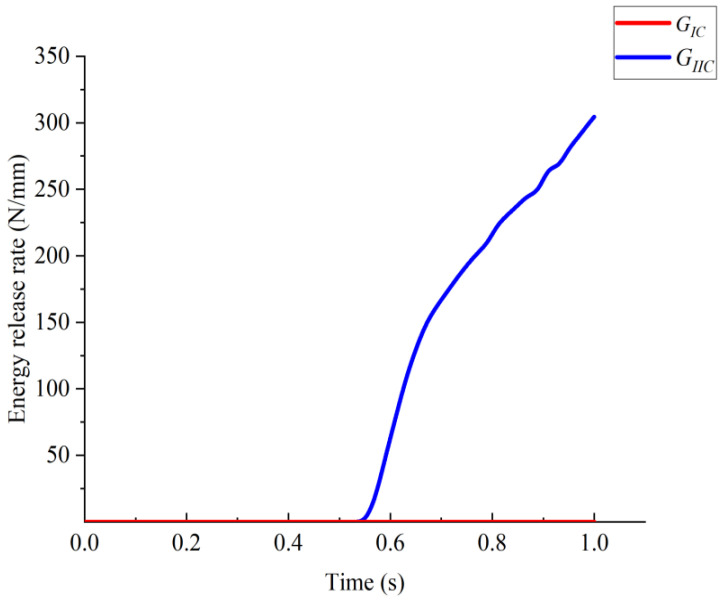
Energy release rate versus time for model II delamination failure.

**Figure 9 materials-15-06880-f009:**
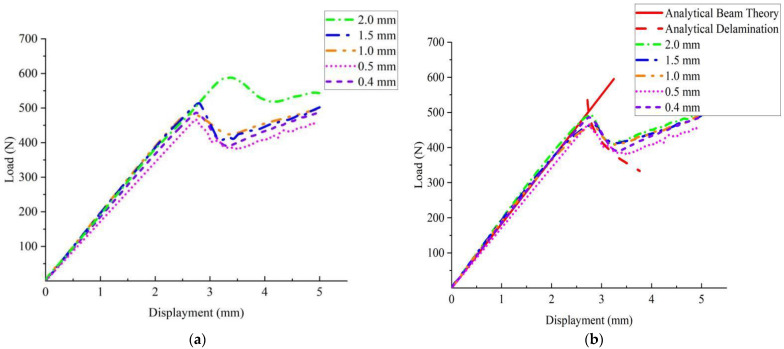
Model II delamination failure based on the tiebreak contact algorithm correction. (**a**). Uncorrected load–displacement curve; (**b**). corrected load–displacement curve.

**Figure 10 materials-15-06880-f010:**
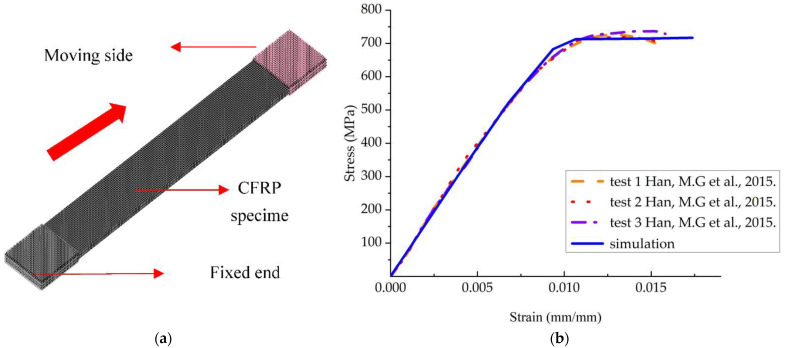
Schematic diagram of nonlinear shear. (**a**) Schematic diagram of the specimen; (**b**) specimen stress–strain curve [28].

**Figure 11 materials-15-06880-f011:**
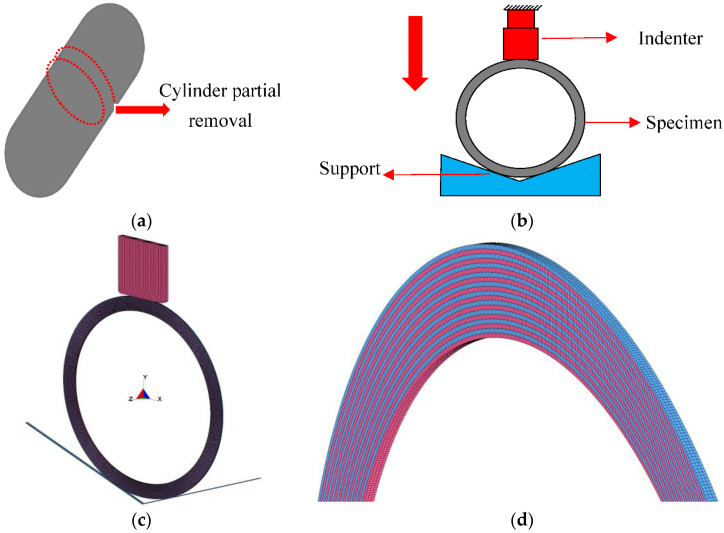
Schematic diagram of simulation analysis for the column part of composite material tank. (**a**) The cutting of the column part of the composite tank; (**b**) the loading experiment on the composite tank; (**c**) finite element modeling of the composite tank; (**d**) the composite tank column elements.

**Figure 12 materials-15-06880-f012:**
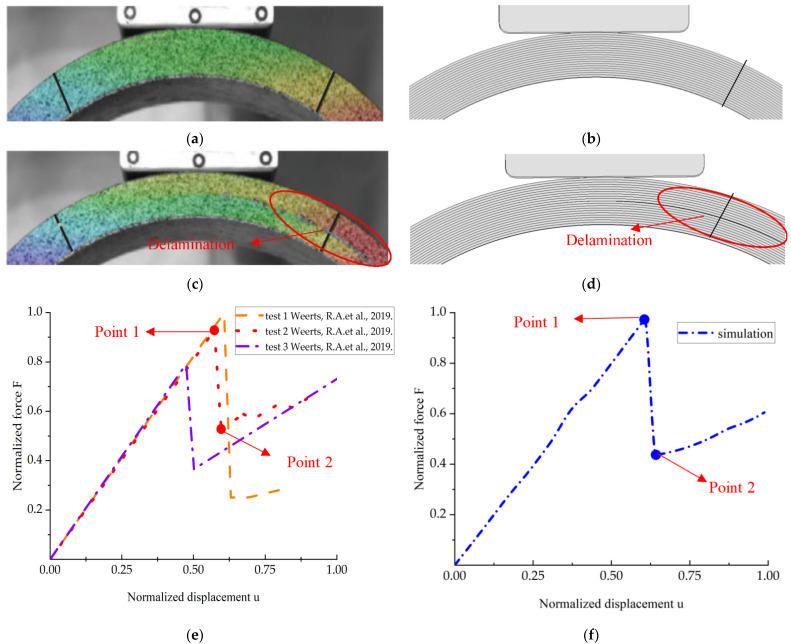
Experiment and simulation of the first delamination failure. (**a**) Experiment point 1 [9]; (**b**) simulation point 1; (**c**) experiment point 2 [9]; (**d**) simulation point 2; (**e**) force–displacement curves of experiment [9]; (**f**) force–displacement curve of simulation. Reprinted/adapted with permission from Ref. [9]. Life of current edition, Houcheng Fang, Di Wang.

**Figure 13 materials-15-06880-f013:**
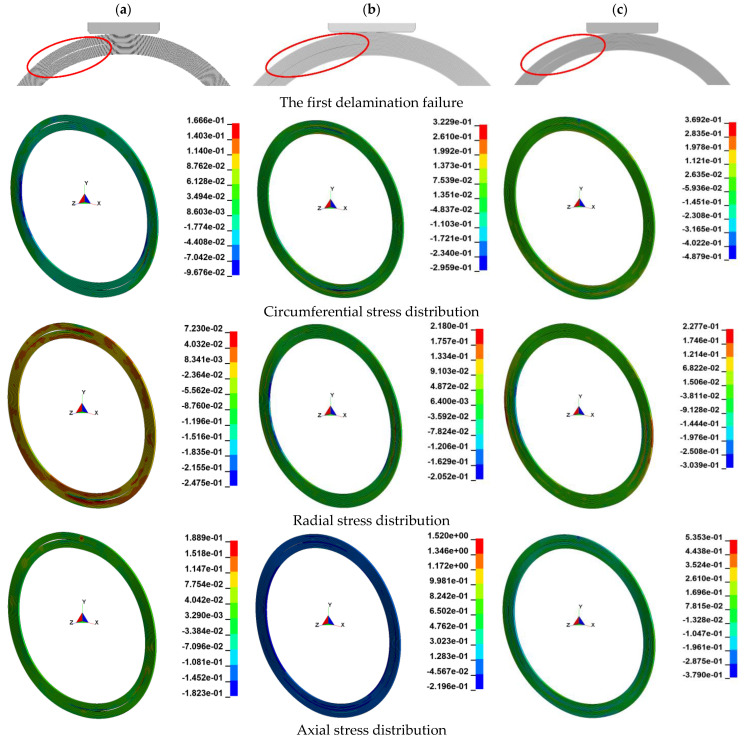
Schematic diagram of the first delamination failure position under different laying methods. (**a**). [0°]_100_; (**b**). [(±45°)_5_ /0°_10_]_5_; (**c**). [±45°]_50s_.

**Table 1 materials-15-06880-t001:** Used material properties to compute the failure envelopes for the CFRP material [9]. Reprinted/adapted with permission from Ref. [9]. Life of current edition, Houcheng Fang, Di Wang.

Part	Young’s Modulus	Poisson’s Ratio	Shear Modulus	Material Strength	
CFRP layers	*E*_1_ = 139 GPa *E*_2_ = 10 GPa *E*_3_ = 10 GPa	*υ*_12_ = 0.26	*G*_12_ = 4.7 GPa	*X_T_* = 2.3255 GPa *X_C_* = 1.0175 GPa *Y_T_* = 0.0623 GPa	*Y_C_* = 0.2537 GPa
		*υ*_13_ = 0.26	*G*_13_ = 3.8 GPa		*S_L_* = 0.0896 GPa
		*υ*_23_ = 0.4	*G*_23_ = 4.7 GPa		*S_T_* = 0.0623 GPa

**Table 2 materials-15-06880-t002:** “Weak” interlayer strength [30].

Parameter	Symbols	Value
Energy release rate	*G**_Ⅰ_**_C_*, *G**_Ⅱ_**_C_*, *G**_Ⅲ_**_C_*	0.001 kN/mm
Peak tractive force	*N*, *S*, *T*	0.025 GPa
Stiffness	*K_N_*, *K_S_*, *K_T_*	100 GPa/mm

**Table 3 materials-15-06880-t003:** Location of first delamination failure under different layups.

Parameter	Ratio of ±45° Layups	Details of the Delamination Location
Single laying method [0°]_100_	0%	Between the eleventh and twelfth CFRP layers
Combined laying method [(±45°)_5_ /0°_10_]_5_	50%	Between the tenth and eleventh CFRP layers
Single laying method [±45°]_50s_	100%	Between the tenth and eleventh CFRP layers

## Data Availability

Not applicable.

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
