# Peer review of "Simulation Analysis of Delamination Damage for the Thick-Walled Composite-Overwrapped Pressure Vessels"

_materials, 2022, doi:10.3390/ma15196880_

Round 1
Reviewer 1 Report
The authors presented problems of the delamination damage occurring in the composite-overwrapped pressure vessels. For composite delamination damage, the model was established using the LS Dyna FEM software. Additionally, for in-plane damage of composite materials, the elastic-plastic process was verified by selecting a progressive damage model with quasi-static nonlinear tensile shear of material and the location of the first occurrence of delamination failure was simulated and analyzed by the Tiebreak contact algorithm.
The article is interesting and fits the profile of the journal. I, for one, have no fundamental objections. I think it will be suitable for publication after corrections.
Strengths
Good graphic illustrations except for Fig. 13. Due to the difficulty of fitting on one page, it should be divided into several separate drawings.
Noticed errors
1. The abstract is far too long and partially suitable for an Introduction chapter. The abstract should only contain a very brief introduction to the article and a summary of the content, as well as a very brief description of the research gap that the authors decided to fill. In my opinion, this should be improved.
2. The Introduction chapter in its current form describes the status of the issue. Both the introduction (the elements of which you do not know why are included in the abstract chapter) and the analysis of the state of the issue (not only a dry specification) with the research gap highlighted here are missing.
3. Chapter Conclusions needs to be expanded with at least a sentence of introduction and conclusions for further research. Presenting only "bare" conclusions is not very professional.
4. The list of cited literature is extremely sloppy. Mysterious designations like [C] or [J] deviate from the current template. In addition, less than 25% of the cited sources are from the last 3 years and this need extending.
Small errors
1. A space should be placed before each left square bracket. Applies to the entire work.
2. Wrong pagination of pages 2/3, and 7/8,
3. Table 2. Wrong KN/mm unit.
4. Between value and unit must be space. Applies to the entire work.
Author Response
Thanks to the reviewer for valuable comments. Please see the attachment.

Reviewer 2 Report
Interesting study and exploration of the topic. The write-up needs some style improvement.
Author Response

(The authors gave the same response as above.)

Reviewer 3 Report
Comments for Authors
This paper explains
The composite-overwrapped pressure vessel is a multi-layer fiber wrapped composite structure. The cohesion zone model (CZM) has been widely used in the simulation of delamination failure of composite structures in recent years. Compared with the cohesive elements model, the Tiebreak contact algorithm based on cohesive contact behavior failure has higher computational efficiency. Although CZM has requirements on the model mesh size, for larger simulation structures, a larger mesh size can be used appropriately to save computation time, when similar results can be achieved by correcting the interlayer strength.
In order to verify the delamination damage occurring in the composite-overwrapped pressure vessels, firstly, for composite delamination damage, the composite laminate model was established using the finite element analysis software Ls-Prepost. The type I and type II delamination failure processes of composite structures were simulated and verified based on the Tiebreak contact algorithm for different mesh sizes, respectively, and the approximate equivalent results were achieved by correcting the inter-ply strength. Then, for in-plane damage of composite materials, the elastic-plastic process is verified by selecting a progressive damage model with quasi-static nonlinear tensile shear of sample specimens as an example. Further, under the purpose of generality and simplicity, the location of the first occurrence of delamination failure was simulated and analyzed by the Tiebreak contact algorithm with a larger mesh size, using quasi-static loading of a 25-mm-thick composite-overwrapped pressure vessel cylindrical section as an example. The results show that delamination occurs at approximately the center, which is in general agreement with the experimentally observed phenomenon. And on this basis, the location of the first significant
delamination phenomenon in composite-overwrapped vessels under different laying methods is predicted. It has a certain foreseeability and reference value for delamination damage of composite pressure vessels.
1. Novelty of this work? As I saw many papers on the same topic
2. Figure 4 the mesh is looking very rough. Did you tried mesh independent test.
3. Did you validate your simulation model with any published literature?
4. Paper looking like technical report only please discuss your results critically in lights of literature.
5. Specimen is made of composite material? If yes add the procedure for its fabrication/preparation to main or supplementary file.
6. Figure 13 the scale bar must be at side and not the surface plot
7. Which mode of failure used for this study mode, II or III? And why?
8. How much the difference between simulation and experimental results
9. Comparison table with research
10. Any supplementary videos for simulation and experiments?
11. Quality of images need improvement
12. Reference need to be updated
Author Response

(The authors gave the same response as above.)

Round 2
Reviewer 3 Report
The Authors have fully addressed all comments and I recommend this paper for possible publication in this Journal.